# *Sarcocornia perennis*: A Salt Substitute in Savory Snacks

**DOI:** 10.3390/foods10123110

**Published:** 2021-12-15

**Authors:** Elsa Clavel-Coibrié, Joana Ride Sales, Aida Moreira da Silva, Maria João Barroca, Isabel Sousa, Anabela Raymundo

**Affiliations:** 1National Institute of Agronomic Sciences for Food and the Environment, AgroSup Dijon, University of Bourgogne Franche-Comté, 21000 Dijon, France; elsa.clavel-coibrie@agrosupdijon.fr; 2LEAF—Linking Landscape, Environment, Agriculture and Food, Instituto Superior de Agronomia, Universidade de Lisboa, Tapada da Ajuda, 1349-017 Lisboa, Portugal; joanasales@isa.ulisboa.pt (J.R.S.); isabelsousa@isa.ulisboa.pt (I.S.); 3Unidade de I&D Química-Física Molecular, Department of Chemistry, University of Coimbra, 3004-535 Coimbra, Portugal; aidams@esac.pt (A.M.d.S.); mjbarroca@esac.pt (M.J.B.); 4Escola Superior Agrária, Instituto Politécnico de Coimbra, 3040-360 Coimbra, Portugal

**Keywords:** crackers, halophyte plants, *Sarcocornia perennis*, antioxidant activity, nutritional composition, sensory analysis

## Abstract

Salt is the primary source of sodium in the human diet, and it is associated with hypertension and increased risk of heart disease and stroke. A growing interest in halophyte plants and food products containing this type of ingredient have been observed, to reduce the NaCl daily intake. In the present work, *Sarcocornia perennis* was incorporated as a food ingredient into crackers to replace the salt (NaCl) and to evaluate its impact on physical properties, water activity, nutritional composition, mineral profile, total phenolic compounds, antioxidant activity, and sensory evaluation. Concentrations of powder dried *S. perennis* from 1 to 10% were tested by replacing the initial salt content and adjusting the flour incorporation to the initial formulation. The incorporation of *S. perennis* had no relevant impact on cracker firmness, but it induced an increase in their crispness. Furthermore, the incorporation of this halophyte originated darker crackers, which was revealed by a decrease of L* and an increase of b* values. In terms of nutritional composition, the incorporation of *S. perennis* leads to the improvement of the snack’s nutritional profile, namely in terms of phenolic compounds, antioxidant activity, and minerals, highlighting the high content in potassium, magnesium, and phosphor. Crackers with a content of 5% of *S. perennis* were sensorily well accepted and this level should be considered the limit of incorporation accepted by the panelists. However, by substituting 1% NaCl for an equal amount of *S. perennis*, it is possible to obtain a 70% reduction in sodium content, which is an important contribution to reducing the overall salt content of the diet.

## 1. Introduction

The land area affected by high concentrations of salts in the soil is increasing day by day. Salinity, the accumulation and effect of excessive ions of Na^+^ and Cl^−^ (salt ions) in the soil, limits crop productivity by severely reducing its growth. This results in a negative impact on food security, economy, and environment. An alternative crop source in global food scarcity is the use of maritime plants, such as halophytes to compensate the demand of food in whole world, especially in the dry regions [1].

Halophyte plants that grow naturally in salt marshes and that represent 1–2% of the flora in the world can cope with saline conditions and can be used for reclaiming salt-affected soil in arid-zone irrigation areas [2,3].

Salicornia and *Sarcocornia* genus (Chenopodiaceae, subfamily Salicornioideae), which are extreme salt-tolerant plants, are two potential halophyte plants to be produced as a vegetable crop, and therefore, with the possibility of cultivation in saline environments worldwide [4,5,6]. Plants such as Salicornia and *Sarcocornia* species present close morphological similarity that make their phenotypic distinction by non-specialists difficult [7]. They are mainly distinguished phenotypically by inflorescence characteristics and life-form (annual and perennial life-forms), but both produce similarly succulent leafless shoots that are highly suitable as a vegetable, resembling green asparagus in the European market. *Sarcocornia*, distributed in Mediterranean and mild-winter Atlantic coast of Europe, the Americas, southern Africa and Australia, includes about 28 species of perennial succulent halophytes with erect to prostrate dwarf shrubs and with flowers exerted and inserted at the same level [8]. In the Iberian Peninsula the species *Sarcocornia perennis* (Miller) have been identified by A.J. Scott, *Sarcocornia fruticosa* (L.) by A.J. Scott, and *Sarcocornia alpini* (Lag.) by Rivas-Martínez et al. as well as other species such as *Sarcocornia hispanica*, *Sarcocornia carinata*, and *Sarcocornia pruinosa*. However, *Sarcocornia perennis* is one of the most abundant and representative salt marsh halophytes of Atlantic coasts and Mediterranean systems [7,9,10,11,12].

*Salicornia* spp. is consumed in northern European countries as fresh salads and pickles, or used in gourmet foods, as garnishes or side dishes for meat and fish [5,13].

Although less studied, *Sarcocornia* spp. produces similar succulent shoots with visually appealing aspect in terms of color, freshness, and particular taste to that of *Salicornia* spp. which can be used for food [5,6]. Recently, *Sarcocornia perennis* was used to produce “green salt” (dry powder) of high quality that can be stored for up to three months at room temperature and at 6 °C for longer [2,4].

The potential use of the aerial parts of Salicornia and *Sarcocornia* species as a vegetable source for human consumption is considered promising, given their high nutritional value in terms of natural minerals and trace elements, including magnesium, sodium, potassium, calcium, manganese, iron, zinc, and chromium, and dietary fiber and bioactive compounds [2,4,5,14,15,16,17]. Different phenolic compounds such as coumarin, phenolic acids, hydroxybenzoic acids and flavonoids, lipids with bioactive properties such as linoleic acid and oleic acid, phytosterols (β-sitosterol and stigmasterol), and pectic polysaccharides present in *Sarcocornia* species confer them important biological properties and health benefits such as being antioxidant, cytoprotective effect against Pb-induced toxicity in HEK 293 cell, and protection of the immune and reproductive systems against toxic chemicals inducers of oxidation reactions [2,18,19,20,21].

The consumption of fortified food is a trend due to the consumer demand for foods with a positive impact on health. Throughout the years, there is a growing demand for healthy, tasty, nutritional, and convenient food such as snacks that respond to an active lifestyle. The snacks have been diversified concerning the shape and the addition of ingredients as cereals/seeds, vegetables, spices, or functional compounds in free or encapsulated form [22]. However, the amounts of sodium chloride in the formulation of a variety of foods such as snacks have a strong contribution to the salt dietary intake. The saltine cracker segment accounted for a major market share during 2016, a result of a widespread increase in demand for healthy convenient snacks. The products with the greatest demand have straight positive effects on health: low cholesterol, unsaturated fats and nearly negligible amount of saturated fats [23]. However, the population is consuming more processed foods, associated with high levels of saturated fats, trans fats, sugars, and salt. Salt is the primary source of sodium and increased consumption of this nutrient is associated with hypertension and increased risk of heart disease and stroke [24]. Thus, the use of dried powdered halophyte plants such as *Sarcocornia* as a salt substitute is an innovative and promising strategy to reduce the ingestion of salt and to produce novel food products with functional and health-beneficial properties such as food additives, antimicrobial agents, beverages, leafy salads, microencapsulated oils, and snacks, among others [25].

Therefore, the present study aims to highlight the nutritional and sensorial potential of powdered dried *S. perennis* as a substitute for salt in savory snacks.

## 2. Materials and Methods

### 2.1. Halophyte Plant Origin

*Sarcocornia perennis (*Figure 1) was provided by the Salina Greens company (Alcochete, Setúbal, Portugal). The drying process was performed according to the method optimized by Barroca et al. [4]. Samples were dried by convective drying at 70 °C, the most appropriate methodology for preserving the health-beneficial properties of the plant. As compared with temperatures ranging between 40 °C and 70 °C, the temperature of 70 °C allows a lower drying time and total color difference and a negligible difference of phenolic content. Barroca et al. [4]. Table 1 presents the biochemical composition of fresh *S. perennis*. 

### 2.2. Other Ingredients

Wheat flour T55 from Espiga (Portugal) was used, with the nutritional composition (g/100 g): Protein: 7, Lipids: 1; Fiber: 2.9, salt: <0.03; Energy: 353 kcal. All the ingredients were purchased from a local market.

### 2.3. Cracker Preparation

Cracker samples were prepared according to the formulations (*w/w*) described in Table 2. Concentrations of powder dried *S. perennis* from 1 to 10% were tested, replacing the initial salt content and by adjusting the flour incorporation (Table 2). 

Crackers were prepared according to the procedure previously optimized by Mota et al. [27], with the following steps: (i) weighing of the ingredients, considering a 100 g batch, corresponding to around 30 crackers; (ii) mixture of ingredients by hand, with an optimized and reproducible process, to obtain a homogeneous cohesive dough; (iii) lamination and molding—the dough was rolled out using a paste machine (Atlas 150, Marcato, Italy), reproducing the extrusion process to obtain a thickness of 1.8 mm. A square mold (38 × 38 mm) was used to cut the dough sheet into pieces, which were then slightly perforated; (iv) cooking process: the crackers were cooked in a forced-air convection oven at 180 °C for 10 min and then dried at 60 °C for 30 min, cooled down at ambient temperature for 10 min, and stored in closed glass containers, protected from the light. 

Physical analyses were performed after 24 h (texture, color and a_w_) and some of the cracker batches were immediately crushed to powder (using an electric mill) and frozen to be used for biochemical composition and antioxidant capacity analyses.

### 2.4. Cracker Analyses

#### 2.4.1. Color Evaluation

The color of the cracker samples was measured instrumentally using a Minolta CR-400 (Minolta, Osaka, Japan) colorimeter with standard illuminant D65 and a visual angle of 2°, according to the method used by Batista et al. (2019) [28], for similar crackers. The results were expressed in terms of L*, lightness (values increasing from 0% to 100%); a*, redness to greenness (60 to −60 positive to negative values, respectively); and b*, yellowness to blueness (60 to −60 positive to negative values, respectively), according to the CIELab system. Chroma, C* (saturation) was also calculated as defined by C* = [(a*^2^ + b*^2^)]1/2. The measurements were conducted under the same light conditions, using a white standard (L* = 94.61, a* = −0.53, and b* = 3.62), under artificial fluorescent light at room temperature, and measurements were replicated ten times for each formulation sample (one measurement per cracker) as well as for the control.

#### 2.4.2. Dimensions

The characteristic dimensions of the crackers were evaluated using a digital caliper model Z22855F (Powerfix, Pulloxhill, Bedfordshire, UK). The individual width (W) and thickness (T) of 10 crackers from each formulation type were measured; spread ratio (W/T) was calculated accordingly. The weight of the cracker samples was also measured, and the corresponding densities calculated (weight (g)/volume (cm^3^)). All these analyses were carried out 24 h after cracker preparation.

#### 2.4.3. Texture Analysis

Instrumental texture analysis was carried out in a TA.XTplus (Stable Micro Systems, Godalming, UK) texturometer. Crackers were submitted to a penetration test, using the following conditions: 3 mm/s probe speed, with a 5 kg load cell, in a controlled room temperature (20 ± 2 °C), according to the procedure optimized by Batista et al. (2019) [27]. Hardness and crispiness were calculated respectively as the force peak (N) and the time needed to reach the maximum peak (s) on the force vs. time curve (texturogram); Hardness corresponds to the maximum force needed to break the cracker and crispiness can be expressed as the time needed to break the cracker [29], i.e., the shorter the time required to break the sample, the crispier it will be.

#### 2.4.4. Sensory Analysis 

Cracker samples with 5% and 10% of *S. perennis*, as well as the control sample, were tested by an untrained sensory analysis panel (*n* = 49, age: 16–62, gender: 33 F, 16 M), in order to assess the product acceptability, compared to the control, and to stablish the limit of incorporation in sensory terms. The cracker samples were evaluated using and hedonic scale, for the following attributes: aspect, color, aroma, flavor, texture, and global assessment (five levels from “very unpleasant” to “very pleasant”). The buying intention was also assessed, from “would not buy at all” to “would definitively buy” (five levels). The assays were conducted in a standardized sensory analysis room, according to the standard EN ISO 8589.

#### 2.4.5. Total Water Content and Water Activity (a_w_)

Water content was determined gravimetrically, using an automatic moisture analyzer (PMB 202, aeADAM, London, UK) at 130 °C until a constant weight was achieved. Water activity (a_w_) was determined using a thermohygrometer (HygroPalm HP23-AW, Rotronic AG, Bassersdorf, Switzerland) at 20 ± 1 °C. Analyses were repeated in triplicate and performed on powdered cracker samples.

#### 2.4.6. Proximate Biochemical Composition

The crackers’ proximate biochemical composition was analyzed on powdered samples, considering total ash content, crude protein, and total fat content, by standard methods. Total ash content was determined gravimetrically by incineration at 550 °C in a muffle furnace, during 24 h. Crude protein was determined using a DUMAS protein/nitrogen analyzer (VELP Scientific NDA 702 DUMAS Nitrogen Analyzer—TCD detector), according to the AOAC 950.36 official method for baked products [30]. The total nitrogen content was determined, and a conversion factor of 6.25 was used to obtain the crude protein content of the crackers. The cracker total fat content was determined following the procedure used for cereals and derived products, described by the Portuguese standard method [31]. All the chemical analyses were performed in triplicate. Regarding carbohydrates, they were determined by the calculation: (100 − [protein + fat + ash + water content]). Energy value was calculated using the conversion factors designated in Annex XIV of Regulation (EU) N° 1169/2011 [32].

#### 2.4.7. Total Phenolics Compounds and Antioxidant Capacity 

The extracts from cracker powder were obtained by the method described by Barreira et al. and Reis et al. [33,34] with some modifications. The powder of different crackers (2 g) was added to 10 mL of ethanol and strongly homogenized with an Ultraturrax homogenizer (IKA Labortechnik T25 Basic, Staufen, Germany) at 8000 rpm for 2 min. The suspension was left at 20 °C, in the dark, for 5 h. Then, the solutions were centrifuged at 6000 rpm, during 10 min and the collected supernatant was stored at 4 °C, protected from light. Regarding the pellet, all the steps were repeated but leaving the second extraction overnight. In the end both supernatants were mixed. After filtration, the samples were dried using a rotavapor. Finally, the concentrated extract was dissolved with dimethyl sulfoxide (DMSO), solvent to obtain a final concentration of 20 mg/mL. All the samples were stored at 4 °C until further analysis.

Total phenolic content was determined according to Mohankumar et al. [35], using the Folin Ciocalteu assay. Briefly, an aliquot of 150 μL of the different extracts was collected and 140 μL of Folin solution were added. This reaction occurred for 3 min, followed by a dilution with 2400 μL of water to each sample tube. Afterwards 300 μL of 1 M Na_2_CO_3_ were added. Samples were incubated for 2 h in a dark environment at room temperature. The absorbance was determined at 725 nm, against a water blank (UNICAM, UV/Vis Spectrometer—UV4).

The total phenolic content was estimated using a calibration curve with gallic acid as a reference, at a concentration ranging between 0 to 200 μg L^−1^. Results were expressed in gallic acid equivalents (mg GAE g^−1^) of dry *S. perennis* powder and crackers.

Antioxidant capacity in the extracts was determined with two methods. The first one is 2,2-Diphenyl-1-picrylhydrazyl (DPPH) radical scavenging assay used by Brand-Williams et al. [36]. The odd electron of nitrogen atom in DPPH is reduced by receiving a hydrogen atom from antioxidants to the corresponding hydrazine. This reaction causes a corresponding change from violet color to pale yellow in the solution [37]. The DPPH assay was performed by mixing 100 μL of sample extract and 3.9 mL of DPPH radical solution. The reaction mixtures were incubated in darkness at room temperature for 40 min and the absorbance was measured at 515 nm. Two blank assays, one without samples and another without reagents, were also performed. Standard calibration curves were made using Trolox standard solutions that were submitted to the same DPPH protocol (0 to 1000 µmol L^−1^).The antioxidant capacity was also accessed by the Ferric Reducing Antioxidant Power (FRAP) method [38]. This method measures the reducing potential of an antioxidant reacting with a ferric tripyridyltriazine (Fe^3+^-TPTZ) complex and producing a colored ferrous tripyridyltriazine (Fe^2+^-TPTZ). Compounds with reducing properties act their action by breaking the chain of free radicals by donating a hydrogen atom [37]. The FRAP assay was performed by mixing 90 µL of sample extract with 2.7 mL of FRAP reagent (TPTZ solution, ferric chloride solution, acetate buffer) and 270 µL of distilled water. The reaction mixtures were incubated at 37 °C for 30 min and the absorbance was measured at 595 nm. Two blank assays, one without samples and another without reagents, were also performed. Standard calibration curves were made using Trolox standard solutions that were submitted to the same FRAP protocol (0 to 700 µmol L^−1^).The antioxidant capacity of the samples obtained from the two methods was expressed in terms of mg of Trolox Equivalent Antioxidant Capacity (TEAC) per gram of dry extract sample. Analyses were repeated in triplicate and performed on powdered cracker samples.

#### 2.4.8. Pigment Composition Determination

The pigment composition was determined by UV-VIS spectroscopy [39]. Extracts were prepared according to the procedure used by Barreira et al. and Reis et al. [33,34], previously described. The absorbance was measured at 470, 648, and 664 nm for respectively carotenoids, chlorophyll a and chlorophyll b. Analyses were repeated in triplicate and performed on powdered cracker samples.

#### 2.4.9. Mineral Composition Determination

For the quantitative determination of all elements (Cu, Na, K, Fe, Ca, Zn, Mn, Mg and P) an acid digestion of the sample (about 0.5 g) was performed, using a mixture of HNO_3_ and HCl (3:1) at 105 °C (with staged heating) in a DigiPrep MS digester (SCP Science, Quebec, QC, Canada). The determination of mineral profile was performed by atomic absorption spectrophotometry and ICP-OES (iCAP 7000 series, Thermo Scientific, Waltham, MA, USA). Analyses were repeated in triplicate and performed on powdered cracker samples [40].

### 2.5. Statistical Analysis

All experiments were repeated at least three times. Statistical analysis of the experimental data was performed using GraphPad Prism 5 software through variance analysis (one-way ANOVA, Student *t*-test), and by the post-hoc Tukey Honestly Significant Difference (HSD) test for identification of differences between three or more groups at a significance level of 95% (*p* < 0.05). All results have been presented as average ± standard deviation. 

## 3. Results

### 3.1. Cracker Color 

Each of the tested crackers showed an appealing appearance (Figure 2). However, higher concentration of *S. perennis* above 10% turned the lamination process impossible to be carried out.

The color parameters, expressed in terms of lightness (L*), redness to greenness (a*), yellowness to blueness (b*) and chroma (C*) are represented in Figure 3.

In terms of lightness, *S. perennis* addition originated darker crackers, reveled by a decrease of L* from 67.18 for the control to 48.76 for the 10% *S. perennis*-enriched cracker. Although crackers with higher concentration have a similar appearance to some commercial products, this browning was not appreciated by the tasters (see Section 3.4 Sensory analysis).

The a* value indicates red-green component of a color, where +a* and −a* indicate red and green values, respectively. The evaluation of this parameter showed a high experimental error, and it was not possible to establish a relationship between a* and the different levels of incorporation of *S. perennis*.

The yellow compounds (b*) significantly increased (*p* < 0.05) with the incorporation of *S. perennis*, showing an increase of yellowness, until 5% of this halophyte. These results agree with the impact of the addition of *S. perennis* referred to in Figure 2. It is observed that the greater the content of pigments (in this case chlorophyll a, chlorophyll b and carotenoids), associated with a greater concentration of halophytes, the greater the darkening of the crackers. The appearance of the yellow-brown color, when the amount of *S. perennis* increased, can be explained by the degradation of pigments, especially chlorophyl, during baking which produces brown-colored compounds such as pheophorbide [41] to which is added the Maillard reaction [42].

A significant increase (*p* > 0.05) of C* value is observed when concentration of *S. perennis* increased in crackers showing an intensification of cracker color as seen in Figure 2. The chromaticity might have been affected by baking, as mentioned for b*.

### 3.2. Cracker Dimensions

Characteristic dimensions of the *S. perennis* crackers are presented in Table 3. In general, incorporation of *S. perennis* did not show a significant difference (*p* > 0.05) in terms of width, thickness, and spread ratio. However, the incorporation of an increasing concentration of *S. perennis* leads to a significant (*p* < 0.05) slight decrease in weight. In addition, the density of the crackers remains constant when *S. perennis* is added, with no significant difference (*p* > 0.05) between samples. This, including the decrease in weight, is related to a slight decrease in cracker volume. These are important quality parameters, as consumers are usually looking for low-weight and less dense crackers [43]. This suggests that gas retention in crackers is not hindered by the presence of *S. perennis*.

### 3.3. Cracker Texture

Texture of crackers is a relevant attribute for the sensory acceptance of the product, as consumers are looking for a product that is relatively crisp [44]. Hardness of crackers with increasing concentrations of *S. perennis* is represented in Figure 4. The response to rupture was highly variable, resulting from the fragility of this type of product, which is expressed by high standard deviation values. For all the samples, no significant difference was observed (*p* > 0.05) in comparison with the control cracker. However, there is a slight tendency towards a reduction in firmness, with the increase in the levels of incorporation of *S. perennis*. Similar results were obtained by Batista et al. [28] for partial substitutions of wheat flour by microalgae biomass. This behavior can be explained by the gradual replacement of part of the gluten-rich wheat flour by *S. perennis*. 

Crispiness of the crackers with increasing concentrations of *S. perennis* was also measured and results are represented in Figure 5. Crispiness was inversely related to the necessary time to break the cracker, therefore a decrease of break time shows an increase of cracker crispiness. 

In general, there is a significant (*p* > 0.05) reduction in the time needed to cause crackers to break, with the incorporation of increasing levels of *S. perennis*, i.e., as increasing amounts of halophyte are incorporated, the structure breaks more quickly, corresponding to an increase in crispness. However, with a high experimental variation, this trend of increasing crispiness can also be explained by the reduction of wheat flour replaced by *S. perennis* that reduces the respective gluten matrix formation and/or starch gelatinization mechanisms such as for Batista et al. [28] when replacing part of the flour with microalgae biomass.

### 3.4. Sensory Analysis 

Sensory analysis was carried out on crackers with 5% and 10% of *S. perennis* and on the control, with a specific aim to select the preferred one for the nutritional analyses, i.e., to establish the maximum level of halophyte incorporation in this type of products. Figure 6 presents the average scores of sensory parameters evaluated by the panel.

The control cracker was the preferred one, in relation to all the evaluated sensorial attributes, with the highest scores, which was in line with expectations for this type of very innovative product. The texture of crackers with the incorporation of 5% and 10% of *S. perennis* obtained similar scores to the control. This is related to the objective texture measurements resulting in similar values for all the cracker samples. However, the cracker with 5% incorporation of *S. perennis* has scores very close to the control and was preferred against the 10% incorporation.

The buying intention of the product was also evaluated by the panel and the results are presented in Figure 7. Here again the control is the preferred buying, with 63% of panelists agreeing to buy the product, in agreement with the sensory parameters results. Regarding the cracker with 5% of *S. perennis* incorporation, 51% of the panelists would agree to buy this product versus 28% for the cracker with 10% of *S. perennis* incorporation, including 16% of panelists who would probably not buy the 10% product and 31% that would not buy the 10% product at all. Consequently, the nutritional analyses and the bioactive evaluation were only carried out on the *S. perennis*-enriched cracker preferred by the panel, which is the one with 5% *S. perennis* incorporation.

### 3.5. Cracker Proximate Biochemical Composition 

Table 4 presents the proximate biochemical composition of crackers with incorporation of 5% of *S. perennis* as well as *S. perennis* pure extract. The analysis of variance was only performed to compare crackers with 5% of *S. perennis* and the control and for the parameters measured, carbohydrates and energy were calculated from the average values from the other determined parameters.

Results of *S. perennis* pure extract for total ash, total fat, crude proteins, and carbohydrates were similar to those found by Barroca et al. [4].

Concerning the crackers, the water content and water activity (a_w_) are important parameters to consider, as they are low-moisture products, and these parameters directly influence their shelf-life, crispiness, and sensory acceptance. The softening of the cracker is observed when the a_w_ is above the value of 0.5 corresponding to the critical a_w_ [45,46].

The water activity of the cracker with 5% of *S. perennis* incorporation did not show a significant difference from the cracker control (*p* > 0.05) and the values were all well below the critical a_w_ with no negative impact on crispiness and sensory acceptance of the product. The a_w_ values are relatively low which is associated with a longer shelf-life, since spoiling bacteria do not have optimal conditions to grow [47].

In terms of water content, the same average values are observed for the cracker with 5% *S. perennis* incorporation and the control cracker.

For total ash, no significant difference (*p* > 0.05) was observed between the sample with 5% of *S. perennis* incorporation and the control cracker.

No significant difference was also observed for total fat content (*p* > 0.05) with respectively 8.7% and 12.7% for the 5% *S. perennis*-enriched cracker and control (Table 4).

Regarding the protein content, crackers with 5% of *S. perennis* incorporation presented significantly higher value (*p* < 0.05) to the cracker control, which results from the higher protein content of halophyte, compared to wheat flour (Section 2.2).

The carbohydrate value of the control cracker and the cracker with 5% of *S. perennis* incorporation remained relatively constant as well as the energy value with 443.9 and 420.7 kcal/100 g for respectively the control and the 5% cracker. 

### 3.6. Cracker Phenolic Content and Antioxidant Capacity

Figure 8 presents the total phenolic content of *S. perennis* pure extract and of the cracker samples. The pure extract of *S. perennis* has a high content in phenolic compounds with 15.79 mg GAE/g DE but this value is lower than the one found by Barroca et al. [4] (27.40 mg GAE/g DE), which is normal for biological materials that have variations according to all sorts of external influences (soil, climate, etc.) as well as growth phase, maturity, etc.

The incorporation of 5% of *S. perennis* into cracker led to a significant increase (*p* < 0.05) of total phenolic content, in relation to the control, up to 3.5 mg GAE/g DE.

Limited scientific information is available on the composition and content of individual phenolic acids and flavonoids present in halophytic plants, particularly *S. perennis*. Nevertheless, Bertin et al. [18] measured the major phenolic acids and flavonoids contained in *Sarcocornia ambigua* samples and found that the major ones were ferulic, caffeic, vanillic, p-coumaric acids, kaempferol, and galangin. Similar results have been highlighted by Costa et al. [19] for *Sarcocornia ambigua* with kaempferol, quercetin, gallic acid, and hydroxybenzoic acid as major phenolic compounds. Phenolic compounds were also determined in *Salicornia herbacea* with the main ones being procatechuic, ferulic, caffeic acids, quercetin and isorhamnetin [48,49].

It is interesting to note that the chemical composition of *Salicornia* and *Sarcocornia* species may change depending on the geographical area where the plants are growing, where climate differences can occur and soil conditions may be different, especially salinity [4,5,50]. The processing conditions also have an impact on the chemical composition of crackers, namely in the availability of phenyl compounds, which during these stages will be subject to complex reactions with other macromolecules, namely proteins and polysaccharides.

The presence of bioactive compounds in *S. perennis* is associated with an antioxidant potential. The antioxidant capacity of cracker samples with incorporation of *S. perennis* was measured by DPPH and FRAP methods, results are presented in Figure 9.

According to the results obtained for the content of phenolic compounds, a high antioxidant capacity is obtained with both methods for the *S. perennis* pure extract with 58.49 mg Trolox/gDE and 60.72 mg Trolox/gDE respectively for the DPPH and FRAP methods. Similar results were obtained on *Sarcocornia ambigua* [4,20] and *Salicornia herbacea* [16]. Higher antioxidant capacity results were observed for FRAP method which means that phenolic compounds reacted more with the ferric ion than the DPPH radical. 

The incorporation of 5% of *S. perennis* led to a significant increase of antioxidant capacity (*p* < 0.05) of cracker sample with the DPPH method in 5 mg Trolox/g DE compared to the control sample. A similar result was obtained with the FRAP method, with an antioxidant capacity almost five times higher for the 5% *S. perennis* cracker.

### 3.7. Pigment Composition

Pigment composition of *S. perennis* pure extract and of the 5% *S. perennis*-enriched cracker is presented on Table 5.

Chlorophyl a was present in higher amount followed by chlorophyl b and then carotenoids. Higher amounts of chlorophyl a and b and of carotenoids were found by Barreira et al. [51] in fresh *S. perennis* plant confirming the loss of pigments during the plantdrying and milling processing. These pigments show antioxidant and immunomodulation activities that can prevent degenerative diseases [52].

### 3.8. Cracker Mineral Composition

The mineral compositions of *S. perennis* pure extract and crackers with 1% and 5% of *S. perennis* are presented in Table 6. Here, the mineral composition of the cracker with 1% of *S. perennis* was also introduced, which corresponds to the replacement of all the NaCl in the original formulation.

As expected, high values of minerals were found in *S. perennis* pure extract in terms of sodium, potassium, calcium, magnesium, phosphor, iron, copper, zinc, and manganese. Similar results were obtained by Barroca et al. [4] for *S. perennis* and by Bertin et al. [18] for *S. ambigua.*

Incorporation of 1% of *S. perennis* in crackers instead of 1% of salt (control) led to a significant decrease of more than 70% of sodium content (*p* < 0.05). For other minerals, crackers with 1% of *S. perennis* incorporation showed no significant difference (*p* > 0.05) from the control.

Incorporation of 5% of *S. perennis* led to a significant increase (*p* < 0.05) of all the minerals except for iron and copper. According to Regulation (European Community) No1969/2011 [32], this cracker can be considered to be “high in potassium, magnesium, phosphor, and iron, copper and manganese” since mineral quantities are higher than 15% of recommended daily values. However, this cracker has very high sodium content compared to the control (42% more). In this context, there will be several ways to optimize the optimum content of *S. perennis* to be incorporated, taking into account the balance between the final sodium content, the desired antioxidant activity, and the possible claims in terms of minerals.

## 4. Conclusions

The incorporation of *S. perennis* in crackers, as a salt substitute (NaCl), was successfully carried out. These snacks presented a crispy texture, which is rather appreciated by consumers as shown by the results of the sensory evaluation.

Crackers with 5% of *S. perennis* (which corresponds to the sensory acceptance limit) presented an increase in total phenolic compounds and antioxidant capacity. The replacement of salt by an equivalent amount (1%) of *S. perennis* promoted a decrease of more than 70% in the sodium content of the cracker. 

Taking into account that the sensory acceptance limit of *S. perennis* was 5% (*w/w*), but for these conditions, the sodium content is high, the final halophyte level can be adjusted, depending on the desired sodium content, in order to not exceed the daily recommendations. Besides salt, the content of other minerals also increased, making the cracker with 5% incorporation of *S. perennis* rich in “potassium, magnesium, phosphor, iron, copper, and manganese”.

These results highlight the potential of *S. perennis* as a promising food ingredient with health benefits, revealed in terms of minerals and antioxidant potential. However, as a salt replacer, if the halophyte is used in the same proportion as salt, the input in sodium is low. Nevertheless, if the addition of halophyte increases, care must be taken to keep the food below a desirable sodium level. 

In future work it will be interesting to incorporate *S. perennis* in other types of salted products to study the impact on physical and nutritional properties in different matrices, with distinct processing conditions.

## Figures and Tables

**Figure 1 foods-10-03110-f001:**
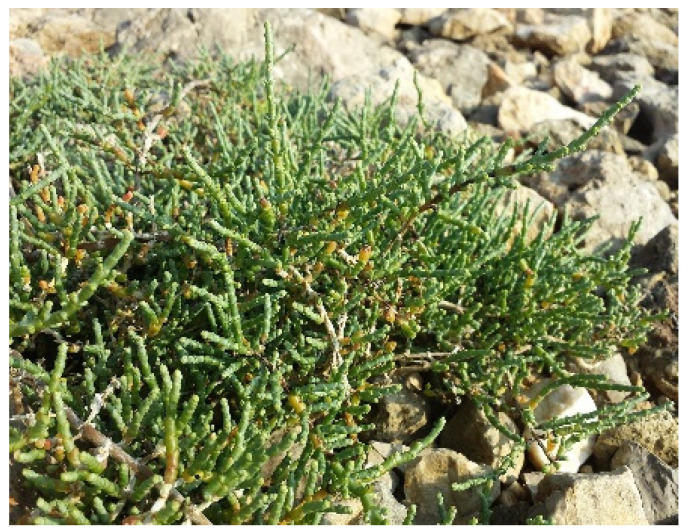
*Sarcocornia perennis* plant [26]. Sarcocornia Perennis Sl17.Jpg—Wikimedia Commons. Available online: https://commons.wikimedia.org/wiki/File:Sarcocornia_perennis_sl17.jpg (accessed on 1 October 2021).

**Figure 2 foods-10-03110-f002:**
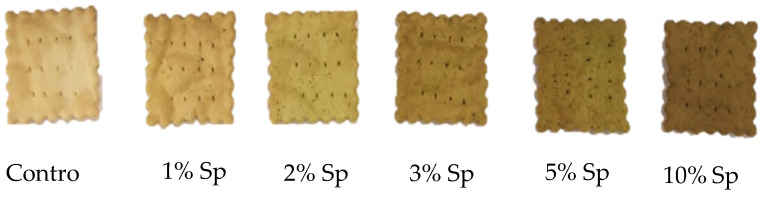
Crackers with incorporation of *Sarcocornia perennis* (Sp) from 1 to 10% (*w/w*) and the control, with 1% of NaCl.

**Figure 3 foods-10-03110-f003:**
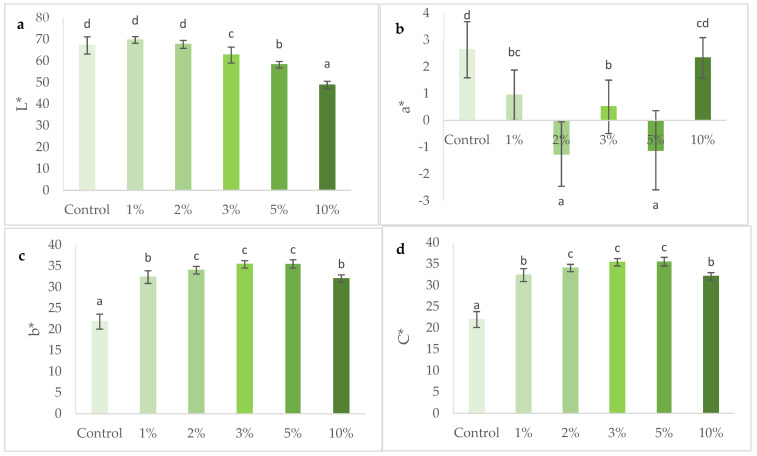
Color parameters lightness (L*) (**a**), greenness (a*) (**b**), yellowness (b*) (**c**) and chroma (C*) (**d**), of crackers with 0% to 10% (*w/w*) *S. perennis* incorporation. Results are expressed as average ± standard deviation (*n* = 10). Different lowercase letters correspond to significant differences (*p* < 0.05) between samples.

**Figure 4 foods-10-03110-f004:**
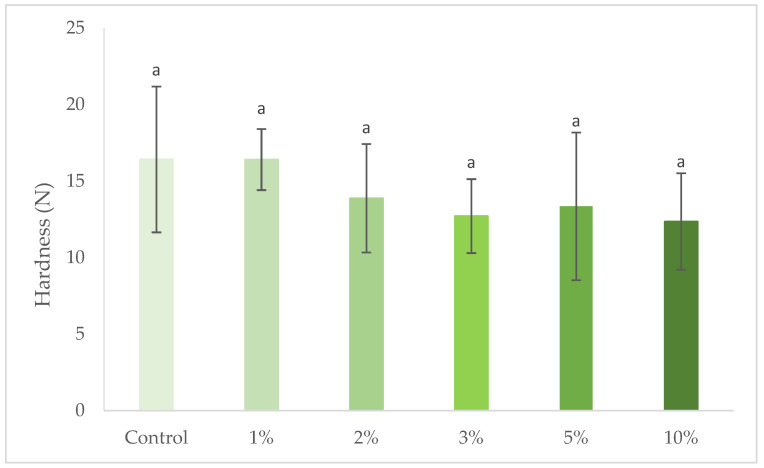
Hardness of crackers with 1% to 10% (*w/w*) *S. perennis* incorporation. Results are expressed as average ± standard deviation (*n* = 10).

**Figure 5 foods-10-03110-f005:**
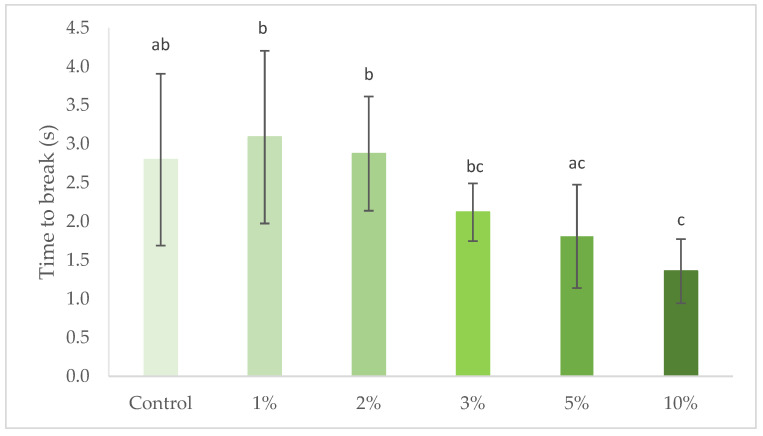
Time to break crackers in a penetration test, at 3 mm/s, with 1% to 10% (*w/w*) *S. perennis* incorporation. Results are expressed as average ± standard deviation (*n* = 10). Different lowercase letters correspond to significant differences (*p* < 0.05) between samples. Results are expressed as average ± standard deviation (*n* = 10).

**Figure 6 foods-10-03110-f006:**
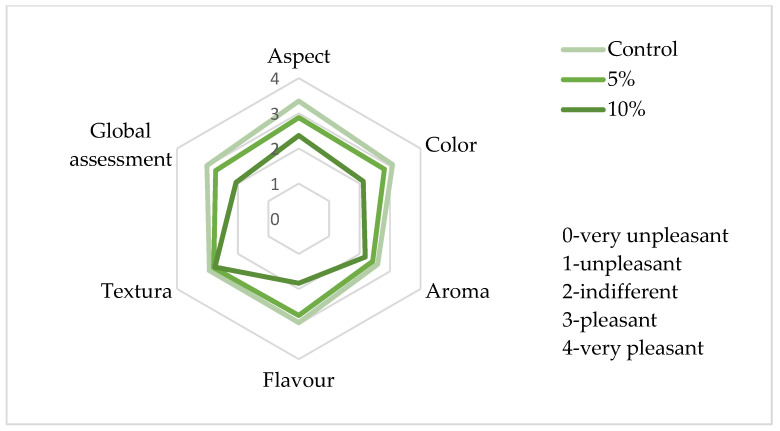
Responses of the sensory analysis panel tasters (*n* = 49) regarding crackers enriched with 5% and 10% of *S. perrennis* as well as the control.

**Figure 7 foods-10-03110-f007:**
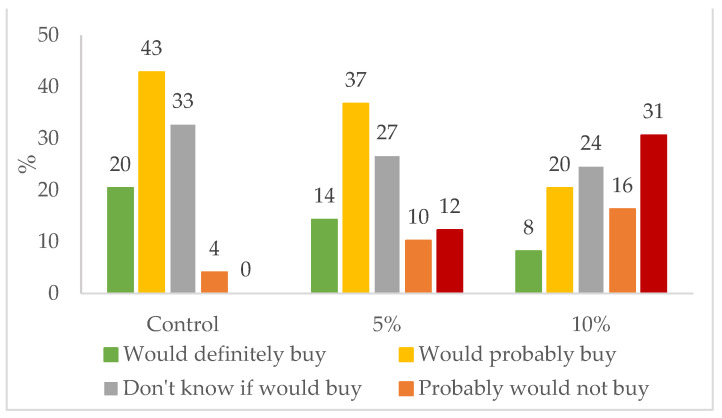
Responses of the sensory analysis panel tasters (*n* = 45) in terms of buying intention for crackers enriched with 5% and 10% of *S. perennis* as well as the control sample.

**Figure 8 foods-10-03110-f008:**
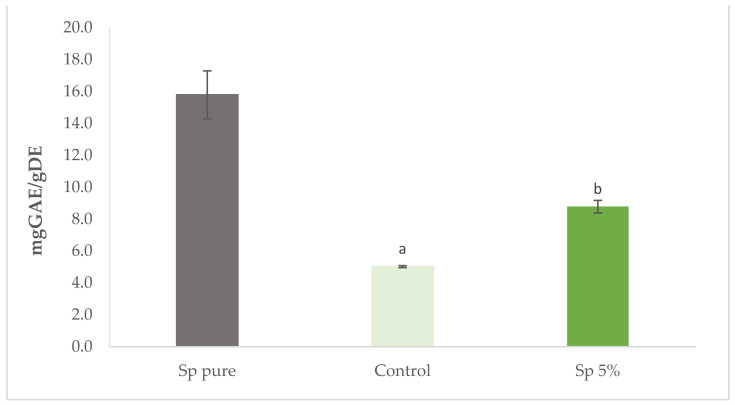
Total phenolic content (expressed as gallic acid equivalents mg·g^−1^ dry weight) of crackers enriched with 5% of *S. perennis* (Sp) incorporation as well as the control and *S. perennis* pure extract. Results are expressed as average ± standard deviation (*n* = 3). Different lowercase letters correspond to significant differences (*p* < 0.05) between samples.

**Figure 9 foods-10-03110-f009:**
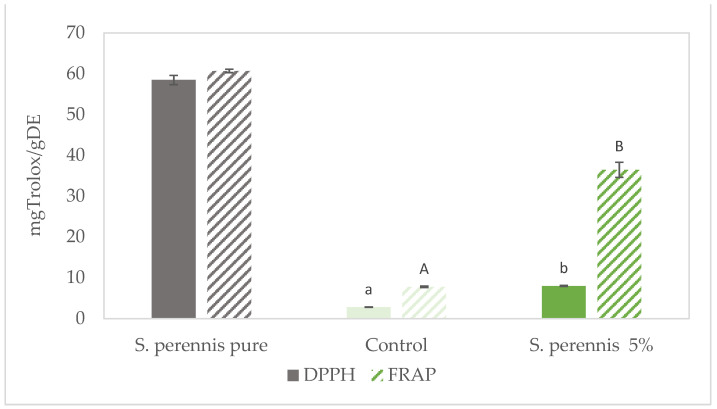
Antioxidant capacity (expressed as mg TEAC/gDETrolox equivalent mg.mg^−1^ dry weigh) of crackers enriched with 5% of *S. perennis* as well as the control and *S. perennis* pure extract. Results are expressed as average ± standard deviation (*n* = 3). Different lowercase letters correspond to significant differences (*p* < 0.05) between samples obtained with DPPH method. Different uppercase letters correspond to significant differences (*p* < 0.05) between samples obtained with FRAP method.

**Table 1 foods-10-03110-t001:** Biochemical composition of fresh *S. perennis* plant [4].

	Fresh *S. perennis* Plant g/100 g
Crude protein	15.60 ± 0.03
Total lipids	1.37 ± 0.07
Crude fiber	8.63 ± 0.15
Ash	43.62 ± 0.05
Carbohydrates	30.80 ± 0.19

**Table 2 foods-10-03110-t002:** Cracker formulations (%, *w/w*). F1—control cracker formulation; F2—1% *S. perennis* cracker formulation; F3—2% *S. perennis* cracker formulation; F4—3% *S. perennis* cracker formulation; F5—5% *S. perennis* cracker formulation; F6—10% *S. perennis* cracker formulation.

Ingredients	F1 (Control) g/100 g	F2 g/100 g	F3 g/100 g	F4 g/100 g	F5 g/100 g	F6 g/100 g
Wheat flour	62	62	61	60	58	57
Sunflower oil	8.5	8.5	8.5	8.5	8.5	8.5
Water	28.5	28.5	28.5	28.5	28.5	28.5
Salt (NaCl)	1	0	0	0	0	0
*S. perennis*	0	1	2	3	5	10

**Table 3 foods-10-03110-t003:** Characteristic dimensions of crackers with 0% to 10% (*w/w*) *S. perennis* incorporation. Results are expressed as average ± standard deviation (*n* = 10). Different lowercase letters correspond to significant differences (*p* < 0.05) between samples.

	Width (W) (mm)	Thickness (T) (mm)	Spread Ratio (W/T)	Weight (g)	Density (g/cm^3^)
Cracker Control	37.4 ± 0.5 ^ab^	4.0 ± 0.8 ^bc^	9.7 ± 1.8 ^a^	2.2 ± 0.1 ^a^	0.41 ± 0.09 ^a^
Cracker with 1% *S. perennis*	37.8 ± 0.8 ^ab^	3.2 ± 0.5 ^ab^	12.2 ± 1.7 ^b^	1.9 ± 0.1 ^bd^	0.43 ± 0.09 ^a^
Cracker with 2% *S. perennis*	38.0 ± 0.6 ^a^	4.5 ± 1.1 ^c^	8.9 ± 2.1 ^a^	2.1 ± 0.1 ^ac^	0.35 ± 0.10 ^a^
Cracker with 3% *S. perennis*	37.9 ± 0.6 ^a^	3.4 ± 0.3 ^ab^	11.3 ± 1.0 ^ab^	2.0 ± 0.1 ^bc^	0.42 ± 0.05 ^a^
Cracker with 5% *S. perennis*	37.1 ± 0.3 ^b^	4.0 ± 0.7 ^bc^	9.6 ± 1.8 ^a^	1.9 ± 0.1 ^b^	0.36 ± 0.07 ^a^
Cracker with 10% *S. perennis*	37.4 ± 0.7 ^ab^	2.9 ± 0.5 ^a^	13.3 ± 2.4 ^b^	1.7 ± 0.1 ^d^	0.44 ± 0.07 ^a^

**Table 4 foods-10-03110-t004:** Proximate biochemical composition (g/100 g) of *S. perennis* (powder), crackers with no additions and crackers with 5% (*w/w*) of *S. perennis* addition. Results are expressed as average ± standard deviation (*n* = 3). Different lowercase letters correspond to significant differences (*p* < 0.05) between samples.

	Water Activity (a_w_)	Water Content (g/100 g)	Total Ash (g/100 g)	Total Fat (g/100 g)	Crude Protein (g/100 g)	Carbohydrates (g/100 g)	Energy Value (Kcal/100 g)
*S. perennis* pure	0.421 ± 0.003	7.6 ± 0.1	33.9 ± 0.1	2.3 ± 0.8	16.7 ± 0.1	39.5	245.5
Cracker control	0.162 ± 0.008 ^a^	3.0 ± 0.1 ^a^	1.9 ± 0.0 ^a^	12.7 ± 5.9 ^a^	9.7 ± 0.0 ^a^	72.7	443.9
Cracker with *S. perennis* 5%	0.159 ± 0.017 ^a^	3.0 ± 1.3 ^a^	2.7 ± 0.6 ^a^	8.7 ± 3.0 ^a^	10.2 ± 0.1 ^b^	75.4	420.7

**Table 5 foods-10-03110-t005:** Pigment composition (mg/gDE) of crackers with 5% of *S. perennis* (*w/w*) as well as *S. perennis* pure extract. Results are expressed as average ± standard deviation (*n* = 3).

	Chlorophyl a (mg/gDE)	Chlorophyl b (mg/gDE)	Carotenoids (mg/gDE)
*S. perennis* pure	0.0365 ± 0.0001	0.0265 ± 0.0007	0.0168 ± 0.0005
Cracker *S. perennis* 5%	0.0021 ± 0.0001	0.0021 ± 0.0002	0.0069 ± 0.0001

**Table 6 foods-10-03110-t006:** Mineral composition (mg/100 g) of crackers with 1% and 5% of *S. perennis*, pure *S. perennis* flour and cracker control. Results are expressed as mean ± standard deviation (*n* = 3). Different letters in the same line correspond to significant differences (*p* < 0.05) between samples.

Mineral (mg/100 g)	15% of the Recommended Daily Values	*S. perennis* Pure	Cracker Control	Cracker *S. perennis* 1%	Cracker *S. perennis* 5%
Na	750	7119.35 ± 195.12	639.93 ± 5.35 ^a^	188.37 ± 3.38 ^b^	908.11 ± 10.60 ^c^
K	300	1830.54 ± 49.15	203.14 ± 2.10 ^a^	204.47 ± 2.39 ^a^	335.18 ± 4.25 ^b^
Ca	120	490.87 ± 10.02	20.89 ± 0.24 ^a^	31.73 ± 2.35 ^b^	58.74 ± 1.32 ^c^
Mg	56.2	786.39 ± 20.94	20.89 ± 0.24 ^a^	37.65 ± 0.55 ^b^	86.38 ± 1.12 ^c^
P	105	230.56 ± 2.32	96.23 ± 0.72 ^a^	96.26 ± 1.7 ^a^	108.65 ± 0.83 ^b^
Fe	2.2	8.64 ± 0.18	2.62 ± 1.69 ^a^	2.78 ± 0.98 ^a^	2.88 ± 0.34 ^a^
Cu	0.2	0.68 ± 0.01	0.24 ± 0.04 ^a^	0.24 ± 0.02 ^a^	0.25 ± 0.00 ^a^
Zn	1.6	2.45 ± 0.03	0.83 ± 0.02 ^a^	0.78 ± 0.02 ^b^	0.96 ± 0.02 ^c^
Mn	0.4	3.49 ± 0.04	0.78 ± 0.01 ^a^	0.80 ± 0.03 ^a^	0.88 ± 0.00 ^b^

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
