# Peer review of "Sarcocornia perennis: A Salt Substitute in Savory Snacks"

_foods, 2021, doi:10.3390/foods10123110_

Round 1
Reviewer 1 Report
The overall content of the manuscript is interesting with rich information about an alternative way of reducing salt in snacks using S. perennis with the end product.
The quality of the writing in terms of the expression and linking of ideas could have been improved. This coupled with grammatical errors means that the manuscript is sometimes hard to follow. Therefore, the manuscript should be carefully checked for clarity and mistakes in the English.
Abstract:
Line 18: add “by” ..were tested by replacing
Lines 18-22: please reword for clarity
Introduction
Lines 45-47: please reword for clarity
Lines 77-78: .. more nutritive foods with health promoting properties? Would read better than with beneficial human health
Lines 83-88: reword for clarity
Line 106: why convective drying at 70 °C is the most appropriate method for preserving the health-beneficial properties of the plant? Please expand on this statement with reference.
Line 106: … by adjusting the flour concentration
Line 178: reword. For example –… and to assess the product acceptability.
Results:
Lines 264-266: reword…For example, Each of the tested crackers showed an appealing appearance ( Figure 2). However, higher concentration of s. perennis above 10% resulted in….
Also the Figure 2 shows that the samples become darker with the increase concentration of SP. Please comments on this and link that to the appealing appearance. Or do you mean that although the samples were darker they were still within the acceptability range?
Line 345: what do you mean by degradation of the pigment. Please reword the whole paragraph for clarity
Line: 389, what do you mean by global terms? In general?
Conclusion: reword for clarity.
Author Response
The overall content of the manuscript is interesting with rich information about an alternative way of reducing salt in snacks using S. perennis with the end product.
The quality of the writing in terms of the expression and linking of ideas could have been improved. This coupled with grammatical errors means that the manuscript is sometimes hard to follow. Therefore, the manuscript should be carefully checked for clarity and mistakes in the English.
We appreciate all the comments related with our work and the positive appreciation. All the work was revised in terms of the English language, presenting an improved version. All changes are highlighted in the text.
Abstract:
Line 18: add “by” ..were tested by replacing
It was added.
Lines 18-22: please reword for clarity
The text was rewritten to make it simpler and clearer.
Introduction:
Lines 45-47: please reword for clarity
The text was rewritten to make it clearer.
Lines 77-78: .. more nutritive foods with health promoting properties? Would read better than with beneficial human health.
The text has been revised.
Lines 83-88: reword for clarity
The text was revised.
Line 106: why convective drying at 70 °C is the most appropriate method for preserving the health-beneficial properties of the plant? Please expand on this statement with reference.
Drying optimization conditions are described in reference:
Barroca, M.J.; Guiné, R.P.F.; Amado, A.M.; Ressurreição, S.; da Silva, A.M.; Marques, M.P.M.; de Carvalho, L.A.E.B. The Drying Process of Sarcocornia Perennis: Impact on Nutritional and Physico-Chemical Properties. J Food Sci Technol 2020, 57, 4443–4458, doi:10.1007/s13197-020-04482-7.
However, more details were added in the text to make the 70°C option clearer.
Line 106: … by adjusting the flour concentration
It was corrected.
Line 178: reword. For example –… and to assess the product acceptability.
The sentence has been clarified. However, in addition to the acceptance of the crackers, compared to the control, it was important to determine the maximum acceptance limit of S. perennis in this type of product.
Results:
Lines 264-266: reword…For example, Each of the tested crackers showed an appealing appearance ( Figure 2). However, higher concentration of S. perennis above 10% resulted in….
The text was rewritten.
Also the Figure 2 shows that the samples become darker with the increase concentration of SP. Please comments on this and link that to the appealing appearance. Or do you mean that although the samples were darker they were still within the acceptability range?
In fact, the cookies became darker, with the incorporation of SP and this had a negative implication in terms of sensory appreciation. This aspect has been clarified in the text.
Line 345: what do you mean by degradation of the pigment. Please reword the whole paragraph for clarity.
The text was clarified.
Line: 389, what do you mean by global terms? In general?
It was replaced.
Conclusion: reword for clarity.
The conclusions were rewritten and clarified.

Reviewer 2 Report
Totally, the research has been well designed. Although, there are some points to review.
Line 205: More details should be added for preparation of extract
Line 213: eliminate this unclear sentence from the text “This method is based on the measurement of the scavenging capacity of antioxidants”
Section 2.4.7 (total phenolics compounds and antioxidant capacity) wrote poor and unclear, should be re-write this section.
Section 2.5: what’s your repetitions? It seems authors carried out the experimental results by duplicate because the standard deviation of the result so high! I strongly recommend repeating the experiments at least triple to decrease STD for further studies.
Section 2.4.1: should be mentioned a reference for the method. Add below reference for this section
Amoli, P.I.; Hadidi, M.; Hasiri, Z.; Rouhafza, A.; Jelyani, A.Z.; Hadian, Z.; Khaneghah, A.M.; Lorenzo, J.M. Incorporation of Low Molecular Weight Chitosan in a Low-Fat Beef Burger: Assessment of Technological Quality and Oxidative Stability. Foods 2021, 10, 1959. https://doi.org/10.3390/foods10081959
Line 326: replace -a* and +a* to positive and negative, please apply for all the text.
Author Response
Totally, the research has been well designed. Although, there are some points to review.
We appreciate the careful review of our work. We made all of the proposed changes, which we believe contributed to improving the final version.
Line 205: More details should be added for preparation of extract.
A description of the preparation of extracts has been included.
Line 213: eliminate this unclear sentence from the text “This method is based on the measurement of the scavenging capacity of antioxidants”.
It was eliminated.
Section 2.5: what’s your repetitions? It seems authors carried out the experimental results by duplicate because the standard deviation of the result so high! I strongly recommend repeating the experiments at least triple to decrease STD for further studies.
All experiments were repeated at least three times; the high values of standard deviation associated with texture measurements are usual in this type of material with high fracturability. This was clarified in the text.
Section 2.4.1: should be mentioned a reference for the method. Add below reference for this section
A reference was added.
Line 326: replace -a* and +a* to positive and negative, please apply for all the text.
It was replaced.

Reviewer 3 Report
Manuscript Draft foods-1472667
SARCOCORNIA PERENNIS AS SALT SUBSTITUTE IN SAVORY SNACKS
Minor revision for acceptance
This manuscript describes the incorporation of Sarcocornia perennis as a NaCL replacer in crackers, but also evaluating the effect on physical properties, nutritional and biochemical profile, antioxidant properties and sensory evaluation. The work was well-developed by the investigators and is inside the scope of this journal.
Results are very interesting and highlights the relevance to analyze the contribution of Sp in antioxidant properties but also considering the effect on sodium content. This aspect was well-covered by this study and strengthened this work. The only thing I missed was the analysis of phenolic content and antioxidant activity in crackers containing 1%Sp. It would be expected that sensory response in 1%Sp crackers are not completely different compared with control sample but could be interesting to include the phenolic content and antioxidant activity data in this work.
In terms of English language, I only suggest in Figure 5 changing word “breack” by “break” in the y-axis.
Author Response
This manuscript describes the incorporation of Sarcocornia perennis as a NaCL replacer in crackers, but also evaluating the effect on physical properties, nutritional and biochemical profile, antioxidant properties and sensory evaluation. The work was well-developed by the investigators and is inside the scope of this journal.
Results are very interesting and highlights the relevance to analyze the contribution of Sp in antioxidant properties but also considering the effect on sodium content. This aspect was well-covered by this study and strengthened this work.
We are grateful for the compliment and believe that this study can contribute to the future use of S. perennis as a substitute for NaCl.
The only thing I missed was the analysis of phenolic content and antioxidant activity in crackers containing 1%Sp. It would be expected that sensory response in 1%Sp crackers are not completely different compared with control sample but could be interesting to include the phenolic content and antioxidant activity data in this work.
In fact, it could be interesting to have the results of phenolic compounds and antioxidant activity also for the sample with 1% of S. perennis. However, these analyses were carried out only for the sample that had greater acceptance by consumers (5% sp).
In terms of English language, I only suggest in Figure 5 changing word “breack” by “break” in the y-axis.
It was corrected.

Round 2
Reviewer 2 Report
The authors have revised the manuscript completely.
Author Response
One more revision of English was carried out, and some inaccuracies were found, which are corrected in the current version. Thank you very much for your contribution.
